# The spatial and social correlates of neighborhood morphology: Evidence from building footprints in five U.S. metropolitan areas

**Noah J. Durst**[1]*, **Esther Sullivan**[2], **Warren C. Jochem**[3]

**1** Noah J. Durst, School of Planning, Design and Construction, Michigan State University, East Lansing, MI, United States of America, **2** Esther Sullivan, Department of Sociology, University of Colorado Denver, Denver, CO, United States of America, **3** Warren C. Jochem, School of Geography and Environmental Science, University of Southampton, Southampton, United Kingdom

* durstnoa@msu.edu

## Abstract

Recent advances in quantitative tools for examining urban morphology enable the development of morphometrics that can characterize the size, shape, and placement of buildings; the relationships between them; and their association with broader patterns of development. Although these methods have the potential to provide substantial insight into the ways in which neighborhood morphology shapes the socioeconomic and demographic characteristics of neighborhoods and communities, this question is largely unexplored. Using building footprints in five of the ten largest U.S. metropolitan areas (Atlanta, Boston, Chicago, Houston, and Los Angeles) and the open-source R package, *foot*, we examine how neighborhood morphology differs across U.S. metropolitan areas and across the urban-exurban landscape. Principal components analysis, unsupervised classification (K-means), and Ordinary Least Squares regression analysis are used to develop a morphological typology of neighborhoods and to examine its association with the spatial, socioeconomic, and demographic characteristics of census tracts. Our findings illustrate substantial variation in the morphology of neighborhoods, both across the five metropolitan areas as well as between central cities, suburbs, and the urban fringe within each metropolitan area. We identify five different types of neighborhoods indicative of different stages of development and distributed unevenly across the urban landscape: these include low-density neighborhoods on the urban fringe; mixed use and high-density residential areas in central cities; and uniform residential neighborhoods in suburban cities. Results from regression analysis illustrate that the prevalence of each of these forms is closely associated with variation in socioeconomic and demographic characteristics such as population density, the prevalence of multifamily housing, and income, race/ethnicity, homeownership, and commuting by car. We conclude by discussing the implications of our findings and suggesting avenues for future research on neighborhood morphology, including ways that it might provide insight into issues such as zoning and land use, housing policy, and residential segregation.

**Data Availability Statement:** The block- and tract-level data used in this study are available at the following repository. R and Python scripts for calculating morphometrics, conducting unsupervised classification, and conducting the

descriptive statistics and regression analysis at the census block and census tract levels are also provided. https://www.openicpsr.org/openicpsr/project/197829/version/V1/view.

**Funding:** N.D. and E.S received the award #2048562 from the National Science Foundation, https://www.nsf.gov/ The funders had no role in study design, data collection and analysis, decision to publish, or preparation of the manuscript.

**Competing interests:** The authors have declared that no competing interests exist.

## Introduction

A decades-long shift in how geographers and planners analyze urban form has emphasized how bottom-up and uncoordinated local decision-making gives rise to large-scale, coordinated, morphological patterns that define the size and shape of cities in predictable ways [1]. Urban morphology–the systematic study of the form and configuration of human settlements with an eye toward uncovering the principles and rules of development and design [2]–has been used for centuries to understand, evaluate, and intervene in urban processes [3]. However, the growth of high-resolution satellite imagery, big data, and new computational tools opens up new avenues to document, evaluate, and monitor urban form. The result has been an increased effort to quantify urban form by identifying the morphological metrics of development [4–7]. Morphological understandings of urban spatial organization and evolution can identify underlying mechanisms and characteristics of urban development, to better plan for and manage increasingly complex urban areas [8]. Drawing on methods from data science, urban morphologists have developed new tools and approaches [9] for characterizing street networks [3,10] as well as the form of buildings [9,11]. New data, tools, and techniques mean researchers are not limited to small case studies which have been common in urban morphology studies. Recent research using building footprints has used morphological analysis to characterize patterns of development at the neighborhood level [7]. For example, Jochem and Tatem use publicly available spatial datasets of building footprints to define their constituent elements (size, shape, and placement of structures) in England, Scotland and Wales and to examine the extent to which typologies of neighborhoods derived from unsupervised classification using building footprint morphometrics align with census-defined classifications for rural and urban areas of various types [7].

We adapt and extend this analysis to the U.S. context to analyze the dimensions and distribution of development inscribed in the *morphology of neighborhoods* in five of the ten largest U.S metropolitan areas and to develop a typology of U.S. neighborhoods based on their morphological characteristics. In doing so, we combine the tools of urban morphology with the theoretical contributions from a vast literature in urban studies, sociology, and planning that has explored how neighborhoods are a key mechanism that structures ecological, political and social outcomes in metro regions. Distinct types of neighborhoods (e.g., suburban enclaves, urban cores, rural districts) vary markedly in the characteristics of their population and the opportunities they provide [12–14]. Little is known, however, about whether the morphological characteristics measured by building footprints align with these pre-existing conceptual understandings of neighborhoods and the characteristics of residents in them. We address this gap in this study by answering three primary research questions: Can neighborhood-level estimates of building morphology be used to create a useful typology of U.S. neighborhoods that maps onto conceptual understandings of urban form? How does neighborhood morphology vary across the country and across central cities, suburban areas, and the urban fringe? Do neighborhoods with distinct building morphologies differ in regard to key socio-demographic characteristics?

We employ the recently developed R *foot* package, a set of open-source tools for calculating morphology metrics for building footprints, which Jochem and Tatem (2021) use to identify the constituent elements of building footprints and settlement patterns across all buildings in Great Britain. Using the *foot* package, we calculate morphometrics summarizing the characteristics of building footprints in census blocks across five major U.S. metropolitan areas with different development and land use histories to examine how the morphology of neighborhoods differs across urban-exurban space and between U.S. metros. We measure neighborhood morphology through physical form, specifically the features of building footprints, including the size, shape, and placement of buildings and their relations to each other. We use unsupervised

classification to identify five primary classes of neighborhoods based on building morphology: these include central-city residential neighborhoods, business and commercial districts, first suburbs, late suburbs, and rural areas. We examine the prevalence and variation of neighborhood types across urban space (from central cities to the urban fringe). Finally, we explore whether and how the physical morphology of neighborhoods corresponds with neighborhood-level spatial and social conditions, including population density, the prevalence of multi-family housing, income, race/ethnicity, homeownership, and commuting by car.

## Background

A wide body of literature in the geographic sciences has focused has sought to use morphological analysis to examine urban phenomena [15–18], including the variegated character of urban development [19] and neighborhood-scale distinctions between settlement types [20]. Yet a large portion of quantitative urban morphological research remains focused on definitions of urban vs non-urban by characterizing rates of urbanization [6], differentiating urbanized and non-urbanized areas [5,21] or distinguishing different degrees of compactness and sprawl [4,22]. Others pay attention to more nuanced variation across urban areas. For example, Xingye et al. (2021) [23] apply multifractal analysis to remote sensed imagery and show how three types of urban clusters (urban core areas, medium-sized urban settlements, and small villages and towns) dominate the urban spatial organization of Beijing.

Our analysis draws on a large literature from the fields of urban studies, sociology, and urban planning that has demonstrated that neighborhoods matter for a range of social, political, and ecological processes and outcomes (see van Ham and Manely 2012 and Sharkey and Faber 2014 for reviews). Our analysis examines how neighborhood morphology maps on to varying spatial and sociodemographic characteristics of place. Fine-grained morphological analysis that distinguishes between neighborhood types can elucidate patterns of development across a broader typology of urban development, including in peri-urban neighborhoods where socially vulnerable populations often reside [20]. The availability of large spatial datasets of building footprint polygons enables more nuanced analysis of variation in the built environment within and across urban areas. Morphology metrics can characterize the size, shape, and placement of buildings and the relationships between them, which can in turn be correlated with or indicative of different neighborhood or settlement types [7].

Morphological analysis using building footprints can identify neighborhood types within single urban areas and classify development patterns across different metropolitan regions. Analysis of urban morphology can provide insight into historical patterns of development, but it requires contextual interpretation [24]. In the U.S. context for instance, the dominant residential building pattern is suburban, as historians of U.S. development have noted [25]. Yet, suburbs are not a monolith. Suburbanization followed multiple waves from the earliest Victorian "first suburbs," to later railroad suburbs, to car-centered suburban sprawl, to "techno-burbs" enabled by contemporary revolutions in communications [26]. Taking suburbs as an example, there are various corresponding economic, demographic, planning, and Census-based definitions of neighborhood types. In a departure from these socioeconomic or regulatory definitions of neighborhood types, a morphological typology of neighborhoods would codify elements of the built environment that distinguish and define the form of U.S. neighborhoods, allowing for more systematic comparison across time and space [27].

## Data and methods

Given the computational intensity of creating building footprint-based measures of neighborhood morphology, in this analysis of neighborhood morphology in the U.S. we focus on a

handful of metropolitan areas. To examine potential variation in morphology across different contexts (urban/rural, older/newer, weakly/strictly regulated), we examine five of the ten largest Combined Metropolitan Statistical Areas (CMSAs) in the country. These five metros represent a range of development and planning histories that are representative of U.S. jurisdictions more broadly. Development patterns are intricately linked to local governments' decisions on how to regulate land, which determines density, the supply and characteristics of buildings, the socio-demographics of populations, the nature of sprawl and the relation of places to the natural environments within and around them [28]. In short, the character of local land use regulations determines the physical character of places in the U.S. These five metros cover all of the four orders that Pendall, Puentes, and Martin (2006) identify as characteristic of U.S. land use regulatory regimes nationally, which they define as: Traditional (Atlanta, Chicago), Exclusionary (Boston), Wild Wild Texas (Houston), and Reform (Los Angeles).

We begin by collecting building footprints for each metropolitan area in question. We use a national database of building footprints generated by Microsoft for more than 125 million buildings in the U.S. The building footprints are two-dimensional representations of the outlines of structures detected in very high-resolution satellite imagery and extracted and mapped using deep neural networks. The building footprint polygons do not contain any additional attribute data which might identify the type of structure. These data were released for public use in 2018 and are publicly available at https://github.com/microsoft/USBuildingFootprints. We then identify all building footprints located within the boundary of the Census Bureau-delineated Combined Statistical Area (CSA) for each of the five metro areas studied. Prior to calculating neighborhood morphometrics, we remove buildings with a footprint of less than 25 meters, which we suspect contain uninhabited structures such as sheds or garages.

We conduct all measurement of neighborhood morphology in R using the *foot* package [7,29] which provides a variety of easy-to-use and flexible options for the calculation of building footprint-derived morphometrics. The building footprints are reprojected into the modal UTM projection for the metropolitan area in question to allow for accurate area and distance calculations. We then use the *foot* package in R to calculate morphometrics for buildings in each census block. Although census blocks are an imperfect proxy for neighborhoods, they are the smallest geography delineated by the U.S. Census Bureau and thus allow for relatively a high-resolution spatial scale that is easily linked to demographic and socioeconomic data on individual communities. Within each census block, we measure a series of morphological characteristics of buildings that we believe are likely to vary across neighborhood contexts in the United States. These include the total area of each footprint (in square meters), the compactness of each footprint, the ratio of building length to equivalent-width, the distance (in meters) to the nearest neighbor, the length of the perimeter of each footprint (in meters), and the footprint's shape index. Where applicable, we estimate both the central tendency (median) and variability (interquartile range) of the morphological characteristics at the census block level. The building footprint-level variables and block-level summary statistics we use to calculate each morphometric are shown in Table 1.

To reduce the influence of outliers within each neighborhood, we calculate the median and interquartile range for each of the variables above within each census block. We also calculate a measure of entropy of the orientation of each footprint, the size of the largest footprint in square meters, the total number of buildings, and the number of buildings per square kilometer. After calculating these morphometrics, we examine descriptive statistics for each morphometric across the five metropolitan areas and across central cities, suburban cities, and areas located along the urban fringe. To do so, we use shapefiles from the U.S. Census Bureau for Census Places to identify all incorporated places within each Combined Metropolitan Statistical Area (CMSA). For each metropolitan area, we treat the one or more incorporated places

**Table 1. Morphometric definitions.**

| | Morphometric | Building Footprint-level Variable | Block-level Summary |
|---|---|---|---|
| Size | | | |
| | area_iqr | Building footprint area in square meters | Interquartile range |
| | perimeter_iqr | Building footprint perimeter length in meters | Interquartile range |
| | area_median | Building footprint area in square meters | Median |
| | perimeter_median | Building footprint perimeter length in meters | Median |
| | area_max | Building footprint area in square meters | Maximum |
| Shape | | | |
| | compact_iqr | Polsby-Popper index | Interquartile range |
| | leqwratio_iqr | Ratio of the longest edge of the building footprint's minimum bounding rectangle to the building's equivalent width | Interquartile range |
| | shape_iqr | Ratio of building footprint area to the area of its minimum bounding circle | Interquartile range |
| | compact_median | Polsby-Popper index | Median |
| | leqwratio_median | Ratio of the longest edge of the building footprint's minimum bounding rectangle to the building's equivalent width | Median |
| | shape_median | Ratio of building footprint area to the area of its minimum bounding circle | Median |
| Placement | | | |
| | nndist_iqr | Distance in meters to the nearest building footprint | Interquartile range |
| | nndist_median | Distance in meters to the nearest building footprint | Median |
| | angle_entropy | Orientation of the building's rotated minimum bounding rectangle | Shannon entropy index |
| | foot_density | Number of building footprints | Footprints per square kilometer |
| | settled_count | Number of building footprints | Sum |

that are named in each metropolitan area as the area's central city (e.g., in Boston, Worcester, and Providence are all named in the Boston CMSA, so we treat them all as central cities). All other incorporated places within the metro area are labeled as suburban cities. Lastly, all areas located outside an incorporated place are labeled as the urban fringe.

We then use principal components analysis (PCA) and unsupervised classification (K-means clustering) to develop a typology of neighborhoods. Prior to the use of PCA, we normalize and standardize the distribution of each variable to reduce the potential influence of outliers and the unit of measurement on the PCA and subsequent unsupervised classification. To normalize each variable, we test the skew of the variables' distribution before and after a series of transformations of the following form, $X^t$ and $X^{1/t}$, where X is the variable of interest and t is a number ranging from 1 to 5. We thus transform each variable by calculating the square through fifth and the square root through the fifth root; in addition, we calculate the natural log. We then select the transformation with the least skewed distribution and then standardize each variable so that it has a mean of 0 and standard deviation of 1. After completing these transformations, we use PCA to conduct dimensionality reduction and to examine whether a limited number of dimensions can be used to represent neighborhood morphology.

We then use unsupervised classification to examine the ability of these morphometrics to identify and describe a typology of neighborhoods and to examine the distribution of these neighborhoods across metropolitan contexts and across the urban, suburban, and rural landscape. We create the classification using only measures of building morphology (i.e., we do not include demographic and socioeconomic variables or other urban features such as road networks as has been used in some prior work [10]). We do so because we are explicitly interested in testing whether neighborhood morphology is associated with variation in socioeconomic and demographic data. We use the K-means algorithm in the Scikit-Learn package in Python

**Table 2. Model performance for various classifiers.**

| Number of Classes | K-means | GMM Diagonal | GMM Spherical | GMM Full | Aggl. Single | Aggl. Complete | Aggl. Average | Aggl. Ward |
|---|---|---|---|---|---|---|---|---|
| 2 | 0.24 (0) | 0.23 (0) | 0.25 (0) | 0.2 (0) | 0.77 (1) | 0.77 (1) | 0.77 (1) | 0.17 (0) |
| 3 | 0.19 (0) | 0.14 (0) | 0.19 (0) | 0.12 (0) | 0.41 (2) | 0.14 (1) | 0.53 (2) | 0.16 (0) |
| 4 | 0.16 (0) | 0.07 (0) | 0.13 (0) | 0.07 (0) | 0.39 (3) | 0.14 (1) | 0.39 (3) | 0.12 (0) |
| 5 | 0.16 (0) | 0.05 (0) | 0.14 (0) | 0.08 (0) | 0.38 (4) | 0.13 (1) | 0.34 (4) | 0.09 (0) |
| 6 | 0.16 (0) | 0.06 (0) | 0.13 (0) | 0.06 (0) | 0.38 (5) | 0.11 (1) | 0.33 (5) | 0.09 (0) |
| 7 | 0.15 (0) | 0.05 (0) | 0.13 (0) | 0.05 (0) | 0.32 (6) | 0.1 (1) | 0.32 (6) | 0.08 (0) |
| 8 | 0.15 (0) | 0.06 (0) | 0.13 (0) | 0.03 (0) | 0.28 (7) | 0.09 (2) | 0.28 (6) | 0.09 (0) |
| 9 | 0.14 (0) | 0.05 (0) | 0.11 (0) | 0.04 (0) | 0.28 (8) | 0.09 (2) | 0.28 (7) | 0.07 (0) |
| 10 | 0.14 (0) | 0.05 (0) | 0.12 (0) | 0.04 (0) | 0.26 (9) | 0.09 (4) | 0.23 (7) | 0.07 (0) |

Notes: This table presents the average silhouette score across all clusters for a given model and pre-specified number of clusters. To illustrate potential over-segmentation, the number of clusters containing fewer than 1% of all observations is shown in parentheses.

[30] to conduct the unsupervised classification. We also test alternative algorithms, including Gaussian mixture models (GMMs) and agglomerative hierarchical methods with various tuning parameters, where applicable: for the GMMs, we evaluate models with spherical, diagonal, and full covariance types, whereas, for the agglomerative approach, we evaluate single, complete, average, and Ward linkages. To compare the results of the classifications across algorithms, we calculate silhouette scores for 2 through 10 clusters for each clustering algorithm. Given the computational intensity of silhouette scores, we use a sample of 10,000 census blocks (1.5% of the more than 630,000 census blocks containing building footprints in the five metro areas) to estimate the average silhouette score. As illustrated in Table 2, although the average silhouette scores are highest for the agglomerative hierarchical models with single, complete, and average linkages, this is due to over-segmentation, leading to (in some cases many) clusters capturing only a fraction of the total observations. These clusters do not, therefore, capture meaningful variation in morphology across the sample. Among the remaining models, the K-means and GMM models with diagonal and spherical covariance structures perform the best, with classifications of 2 and 3 classes producing the highest silhouette scores (.2 to .25). Given its comparable performance and its ubiquity in the literature, we select the K-means results for further analysis.

To select the optimal number of clusters, we examine an elbow plot and descriptive statistics for each class from the various K-means models with between 2 and 10 clusters. As shown in the elbow plot (Fig 1), there is no inflection point indicating a clearly optimal model. However, upon subsequent review of the descriptive statistics for the morphometrics, disaggregated by each class (results not shown), it appears that the results from 2- and 3-way classifications primarily distinguish between 1) census blocks with low-density development, 2) high-density development with large buildings, and 3) high-density development with small to moderate sized buildings (e.g., residential neighborhoods). They do not, however, provide much insight into variation within these classes. Given that evaluating variation in the morphology of residential areas is one of the primary objectives of the study, we choose to discuss the results of the classification with 5 clusters because it has the next highest silhouette score and results in multiple classes of low-density, primarily residential development.

We do not claim the 5 classes discussed below represent mutually exclusive or universal neighborhood types. Rather, we describe how these classes differ regarding key morphological characteristics that correspond with broad archetypes in the social science of urban and suburban neighborhoods in the United States. A key contribution of this analysis is our test of

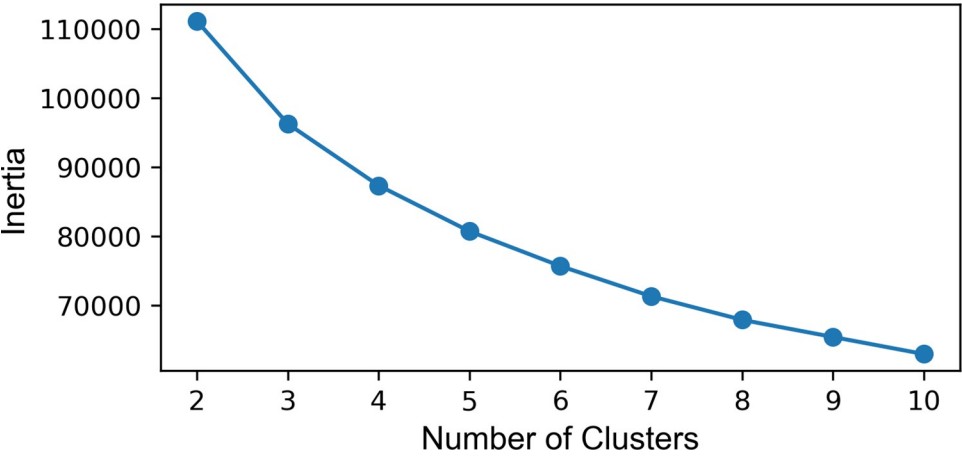

**Fig 1. Elbow plot.**

whether and how neighborhood morphology aligns with socio-demographic characteristics of these archetypes. Selecting a different number of clusters or a different clustering algorithm may lead to neighborhoods with more or less refined and distinct morphological characteristics. But, as we describe below, morphology would still likely correlate with socioeconomic and demographic conditions in ways that map intuitively onto sociological understandings of urban and suburban spaces.

We examine the results from the unsupervised classification by discussing descriptive statistics for the morphometrics for each class and examining the distribution of each class across the five metros and across central cities, suburban cities, and the urban fringe. We then use Ordinary Least Squares regression analysis to examine the relationship between demographic and socioeconomic characteristics and the prevalence of each class at the census tract level. The purpose here is to examine whether the morphology-based classifications map onto social variables in meaningful and informative ways. To do so, we estimate the following regression model:

$$Y_{ij} = \alpha + \beta X_{ij} + \delta D_j + \varepsilon_{ij}$$

where $Y$ represents the share of census blocks within each census tract $i$ and metro area $j$ that are assigned to each of the five morphological classes from the K-means classification; $\alpha$ represents the intercept; $X$ represents a vector of socioeconomic and demographic characteristics for the $i$th census tract in the $j$th metro area, including the median year structures were built, the population density per square mile, the percentage of housing units located in structures with 20 or more units, the homeownership rate, the median household income, the percentage of people who are non-Hispanic White, and the percentage of workers who commute by car; $\beta$ represents a corresponding vector of coefficients that capture the relationship between each socioeconomic and demographic indicator included in $X$; $D$ represents a vector of dummy variables for each metropolitan area; $\delta$ is a vector of coefficients associated with the metropolitan dummy variables and represents the average difference in the share of neighborhoods of each class relative to the reference category (Atlanta), holding other variables constant; and $\varepsilon$ is the error term.

## Results

### Descriptive statistics

As illustrated in Table 3, which shows the median for each morphometric for census blocks in each metropolitan area, the five metropolitan areas differ in regard to the size and placement

**Table 3. Median morphometrics by metropolitan area.**

| | | Atlanta | Boston | Chicago | Houston | Los Angeles |
|---|---|---|---|---|---|---|
| Size | | | | | | |
| | area_iqr | 87.50 | 63.39 | 74.37 | 86.52 | 84.45 |
| | perimeter_iqr | 15.32 | 13.42 | 15.32 | 16.25 | 15.93 |
| | area_median | 192.47 | 146.79 | 157.58 | 195.58 | 213.27 |
| | perimeter_median | 58.21 | 51.04 | 53.03 | 58.70 | 62.73 |
| | area_max | 405.41 | 305.45 | 338.95 | 397.99 | 415.05 |
| Shape | | | | | | |
| | compact_iqr | 0.09 | 0.09 | 0.08 | 0.11 | 0.10 |
| | leqwratio_iqr | 0.53 | 0.55 | 0.53 | 0.53 | 0.54 |
| | shape_iqr | 0.08 | 0.09 | 0.08 | 0.10 | 0.09 |
| | compact_median | 0.72 | 0.72 | 0.72 | 0.73 | 0.70 |
| | leqwratio_median | 1.61 | 1.60 | 1.56 | 1.53 | 1.57 |
| | shape_median | 0.55 | 0.56 | 0.56 | 0.56 | 0.55 |
| Placement | | | | | | |
| | nndist_iqr | 10.49 | 7.85 | 4.55 | 5.95 | 3.33 |
| | nndist_median | 33.05 | 27.30 | 19.90 | 21.79 | 18.46 |
| | angle_entropy | 0.85 | 0.86 | 0.97 | 0.91 | 0.92 |
| | foot_density | 178.77 | 439.76 | 698.02 | 502.56 | 871.61 |
| | settled_count | 15 | 15 | 15 | 17 | 22 |

of buildings in the typical neighborhood, but not in regard to the shape of buildings. For example, the typical size of building footprints in each neighborhood (area_median) and the variability among building footprints within neighborhoods (area_iqr), both differ considerably across metro areas. In Atlanta, Houston, and Los Angeles—the three post-car metros—the median building in the median neighborhood is considerably larger (between 192 and 213 square meters) than in Boston or Chicago (147 to 158 square meters). Similarly, the variability in the size of buildings is also larger in these post-car metros, where in the median neighborhood buildings varied in size with an interquartile range of 84 square meters or more; this is notably more intra-neighborhood variation in building size than is found in Boston (63) or Chicago (74). Thus, neighborhoods in the older metros are typically composed of smaller and more uniformly sized buildings than those in Sunbelt cities.

As illustrated in Table 3, there is also considerable variation between metropolitan areas in regard to the distance between buildings and, relatedly, the number of buildings per square kilometer. For example, in Los Angeles and Chicago–two of the most densely settled metropolitan areas in the country–more than half of buildings in the median neighborhood are within approximately 19 meters of another building. However, in the typical neighborhood in less densely settled Atlanta, most buildings are 33 meters from the nearest building. To put it differently, the building density in Chicago (698 buildings per square kilometer) and Los Angeles (872) is considerably higher than in Atlanta (179). Table 3 also reveals some counter-intuitive and notable findings regarding the morphology of neighborhoods across the five metropolitan areas. For example, in Boston, the distance between buildings (27 meters) is considerably larger than in the post-car metros of Houston (21) and Los Angeles (19). One might expect Boston to have higher building density given its period of development. As we explore in Tables 4 and 5 below, this is largely explained by the location of buildings across central cities, suburban cities, or the urban fringe within each metropolitan area. Similarly, it is notable that building size is not directly related to either building density or distance between buildings.

**Table 4. Median morphometrics by location.**

| | | Central Cities | Suburban Cities | Urban Fringe |
|---|---|---|---|---|
| Size | | | | |
| | area_iqr | 84.29 | 66.37 | 87.10 |
| | perimeter_iqr | 17.64 | 12.80 | 16.76 |
| | area_median | 160.90 | 183.85 | 178.87 |
| | perimeter_median | 53.76 | 57.34 | 56.77 |
| | area_max | 459.08 | 339.29 | 387.48 |
| Shape | | | | |
| | compact_iqr | 0.10 | 0.08 | 0.10 |
| | leqwratio_iqr | 0.64 | 0.46 | 0.58 |
| | shape_iqr | 0.10 | 0.08 | 0.09 |
| | compact_median | 0.71 | 0.72 | 0.71 |
| | leqwratio_median | 1.65 | 1.53 | 1.60 |
| | shape_median | 0.55 | 0.56 | 0.55 |
| **Placement** | | | | |
| | nndist_iqr | 3.57 | 3.76 | 10.15 |
| | nndist_median | 15.75 | 19.35 | 30.28 |
| | angle_entropy | 0.93 | 0.93 | 0.90 |
| | foot_density | 1083.50 | 804.95 | 179.20 |
| | settled_count | 18 | 17 | 16 |

For example, although the post-care metros of Atlanta, Houston, and Los Angeles all have larger buildings (median about 192 square meters), they vary markedly in both building footprint density and distance between buildings. These findings point toward potentially divergent building development patterns within each metropolitan area.

**Table 5. Selected median morphometrics by metropolitan area and location.**

| | | area_median | nndist_median | foot_density | Percentage of Blocks |
|---|---|---|---|---|---|
| Atlanta | | | | | |
| | Central Cities | 187.46 | 21.12 | 492.96 | 5% |
| | Suburban Cities | 197.02 | 28.25 | 382 | 27% |
| | Urban Fringe | 191.17 | 36.35 | 85.61 | 68% |
| Boston | | | | | |
| | Central Cities | 149.2 | 16.62 | 1107.15 | 6% |
| | Suburban Cities | 140.66 | 20.32 | 857.07 | 30% |
| | Urban Fringe | 149.89 | 31.6 | 235.38 | 64% |
| Chicago | | | | | |
| | Central Cities | 128.79 | 11.94 | 1440.77 | 18% |
| | Suburban Cities | 151.79 | 18.37 | 841.11 | 49% |
| | Urban Fringe | 189.99 | 28.32 | 131.98 | 32% |
| Houston | | | | | |
| | Central Cities | 194.14 | 18.81 | 802.54 | 23% |
| | Suburban Cities | 196.85 | 21.14 | 609.02 | 30% |
| | Urban Fringe | 195.44 | 26.67 | 211.54 | 47% |
| Los Angeles | | | | | |
| | Central Cities | 181.15 | 15.67 | 1208.67 | 15% |
| | Suburban Cities | 225.21 | 18.2 | 947.25 | 56% |
| | Urban Fringe | 200.66 | 22.81 | 358.41 | 29% |

To explore variation in building morphology within metropolitan areas, we now turn to an examination across central cities, suburban cities, and the urban fringe, as shown in Table 4. A number of morphometrics show notable variation across these spatial scales. As might be expected, the median footprint of buildings in suburban cities and the urban fringe is considerably larger than in central cities (median of approximately 180 compared with 161). Similarly, buildings in the fringe are much farther from each other (median distance of 30 meters) when compared with buildings in suburban and central cities (19 and 16 meters, respectively), and neighborhoods along the fringe have considerably lower building density (179 buildings per square kilometer) than in central and suburban cities (1,083 and 805 buildings per square kilometer). Notably, however, as illustrated by the interquartile range (IQR) morphometrics, the location with the least intra-neighborhood variability is suburban cities. For example, in suburban cities the typical neighborhood has considerably less intra-neighborhood variation in building size, as indicated by an interquartile range of 66 square meters, compared with 84 and 87 square meters in central cities and the suburban fringe. Suburban cities also show lower intra-neighborhood variation across the other metrics studied here (area_iqr, compact_iqr, leqwratio_iqr, nndist_iqr, perimeter_iqr, and shape_iqr) than do neighborhoods on the urban fringe. The lower variability in suburban morphometrics across very different U.S. metros reflects not only the prevalence of cookie-cutter style suburban neighborhoods with uniform housing types, but also the dominance of common land use regulations and development practices (i.e., setbacks and minimum lot sizes) that shape suburban development patterns.

Table 5 presents selected morphometrics–the median and maximum area, median distance between buildings, and the building footprint density–in each metropolitan area, disaggregated by location within the central city, suburban city, and urban fringe. For comparison across metro areas, we also included the percentage of census blocks in each location. A number of these findings are notable. For example, although in all cases building density decreases (and distance between the nearest building increases) as one moves from central cities to suburban cities and from suburban cities to the urban fringe, the five metropolitan areas differ substantially in regard to the intensity of development across these three locations. For example, in Boston, nearly two-thirds (64%) of census blocks are located in the urban fringe, where the distance between neighboring buildings is 31 meters (second only to the urban fringe of Atlanta). This is driven by the prevalence of low-density, unincorporated New England towns, many of which rely on exclusionary zoning to limit the density of new development [28]. In comparison, in Los Angeles, more than two-thirds of census blocks are in suburban cities (56%) and central cities (15%) and have the highest building footprint density and lowest distance between neighborhoods observed in suburban and fringe areas across the five metropolitan areas.

A second notable finding is that, in some metros, there is minimal if any variation in the size of the median building, while in others there is substantial variation between central city, suburban city, and urban fringe locations. For example, in Atlanta, there is only a 10-square meter difference between the size of the median building in suburban cities (197) and central cities (187). The same is true in Boston (140 to 149) and Houston (194 to 196). In Chicago and Los Angeles, however, the median building in central cities (128 and 181 square meters, respectively) is more than 40 square meters smaller than buildings located in other parts of the metropolitan area. This suggests highly divergent development patterns in these two metropolitan areas wherein suburban cities (in Los Angeles) or the urban fringe (in Chicago) are home to substantially larger buildings than the central city. The results in Chicago make some intuitive sense: buildings in lower-density areas typically have larger footprints; thus, the urban fringe has larger building footprints than suburban cities (189 vs 151 square meters), which in

turn have larger footprints than central cities (128). In Los Angeles, however, suburban cities have substantially larger buildings than exurban areas and central cities (225 vs. 200 and 181, respectively). This is likely driven by what has been called "horizontal density"–the expansion of single-family units and the widespread creation of accessory dwelling units across what were historically exclusively single-family suburban neighborhoods [31,32].

### Unsupervised classification

We now turn to a discussion of the results of our unsupervised classification (K-means using 5 classes). Fig 2 provides archetypal examples of each of the five classes, while Table 6 presents

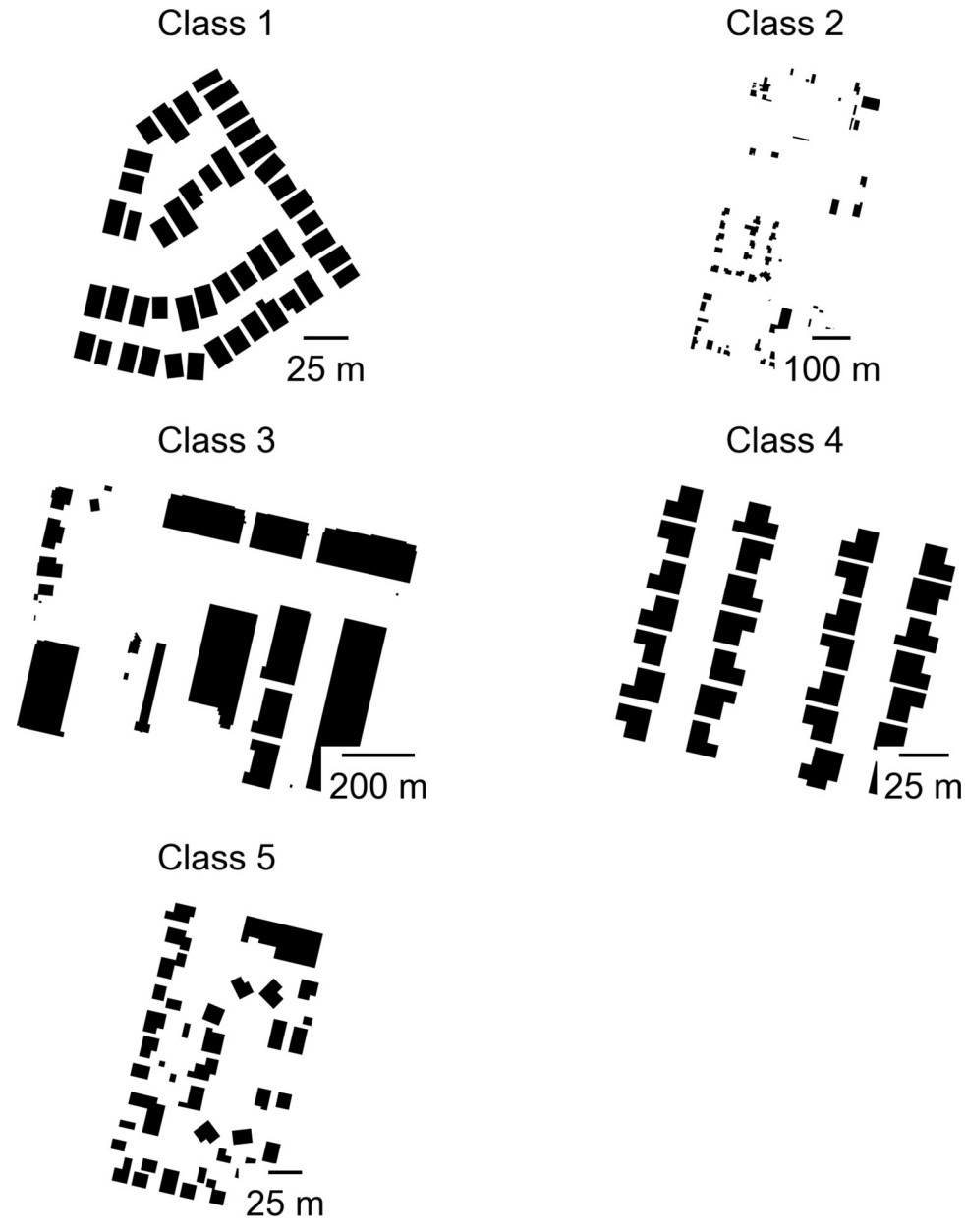

**Fig 2. Archetypal examples of each class.**

**Table 6. Median morphometrics by neighborhood class.**

| | | Class 1 | Class 2 | Class 3 | Class 4 | Class 5 |
|---|---|---|---|---|---|---|
| Size | | | | | | |
| | *area_iqr* | *56.66* | *153.5* | *1140.4* | *63.78* | *95.49* |
| | *perimeter_iqr* | *10.51* | *25.5* | *88.39* | *10.42* | *19.3* |
| | area_median | 157.78 | 198.64 | 863.98 | 246.41 | 158.26 |
| | perimeter_median | 50.72 | 59.46 | 124.14 | 67.06 | 53.08 |
| | area_max | 398.76 | 1438.8 | 5056.22 | 458.57 | 654.16 |
| Shape | | | | | | |
| | *compact_iqr* | *0.05* | *0.12* | *0.17* | *0.07* | *0.12* |
| | *leqwratio_iqr* | *0.33* | *0.77* | *1.3* | *0.4* | *0.79* |
| | *shape_iqr* | *0.06* | *0.12* | *0.15* | *0.06* | *0.12* |
| | compact_median | 0.76 | 0.71 | 0.61 | 0.67 | 0.71 |
| | leqwratio_median | 1.37 | 1.66 | 2.3 | 1.79 | 1.68 |
| | shape_median | 0.6 | 0.54 | 0.47 | 0.52 | 0.54 |
| Placement | | | | | | |
| | *nndist_iqr* | *4.95* | *19.63* | *15.43* | *5.18* | *4.24* |
| | nndist_median | 20.19 | 36.26 | 40.76 | 27.52 | 16.07 |
| | angle_entropy | 0.86 | 0.77 | 0.92 | 0.91 | 0.85 |
| | foot_density | 928.67 | 162.52 | 240.33 | 616.4 | 1262.66 |
| | settled_count | 24.17 | 35.51 | 13.38 | 16.74 | 30.44 |

the median for each of the 16 morphometrics in each of the 5 classes. As is clear, class 1 is primarily composed of neighborhoods with smaller buildings, high levels of building density, and low intra-neighborhood variability in building size, shape, and placement. In other words, these are dense neighborhoods of smaller buildings that vary little from each other in regard to the orientation of buildings. These are likely cookie-cutter, single-family, residential neighborhoods with modestly sized homes. Class 4 is similar, with little intra-neighborhood variation in the size, shape, and placement of buildings, but with lower density and larger buildings (see below). Class 2 on the other hand contains neighborhoods with low overall building density and a high degree of intra-neighborhood variability in regard to building size, shape, and placement. These are therefore low-density neighborhoods which, as we illustrate shortly, are primarily located on the urban fringe. Class 3 is composed primarily of neighborhoods with large buildings with non-compact shapes. These census blocks likely contain commercial or mixed-use buildings or other buildings with large and varied footprints. Lastly, class 5 is characterized by the high density and high variability of building footprints.

Comparisons across the five classes reveal a number of interesting similarities and differences. For example, class 4 is similar to class 1, with low intra-neighborhood variability in building size, shape, and placement, but larger building footprints and lower density. Thus, class 1 may capture earlier suburban developments with modest homes on smaller lots while class 4 may capture more recent suburban-style developments with larger houses on larger lots. Moreover, class 5 is similar to both class 1 and class 4 in regard to the size and shape of the median building, but neighborhoods in class 5 tend to have substantially higher intra-neighborhood variability in building size and shape. In other words, the size and shape of buildings within the same neighborhood vary considerably in class 5 but are relatively uniform in classes 1 and 4. This variability is clearly illustrated in Fig 2, which depicts representative arrangements of building footprints for each neighborhood class. Notably, neighborhoods in class 5 also have substantially higher building density and substantially lower distances

between buildings. Class 5 may therefore represent downtown" or "main street" neighborhoods where there is a greater mix and density of buildings or denser, single-family neighborhoods with weak or weakly enforced land use regulations.

Lastly, there are also interesting similarities between classes 2 and 5. Despite the relatively small size of buildings in both classes, there is a high degree of intra-neighborhood variability in building size and shape in both class 2 and class 5. The primary factor that distinguishes these two classes is the distance between buildings and the overall density of buildings within the neighborhood. Unlike class 5, which has the highest density of all 5 classes (1,262 buildings per square kilometer), class 2 has the lowest building density with a median of 162 buildings per square kilometer.

## Spatial and socioeconomic analyses

Morphological analysis of building footprints alone is clearly able to distinguish a typology of U.S. neighborhoods, but how does this morphology-based taxonomy map onto variation in spatial and social dimensions between neighborhoods? We conclude by examining the distribution of each class across space and the association of each class with key demographic and socioeconomic characteristics. To do so, we examine the share of each class that is located in each metropolitan area and in three sub-metropolitan regions (central cities, suburban cities, and the urban fringe). We also use regression analysis to examine the association between the share of neighborhoods (census blocks) in each tract that were predicted to be of each class and key socioeconomic and demographic data, as measured by 2016–2020 tract-level estimates from the American Community Survey.

We begin by discussing the results for class 3. Recall that, as illustrated in Table 3, class 3 neighborhoods have substantially larger buildings than the other four classes. Table 7 shows that a relatively small share of neighborhoods in each metro area and each sub-metropolitan region are in class 3. For example, class 3 neighborhoods make up a low of 6% of census blocks in Boston and a high of 12% of census blocks in Los Angeles. Similarly, class 3 neighborhoods make up a maximum of 14% of census blocks in central cities, and between 8–9% in suburban cities and the urban fringe. The regression results in Table 8 provide additional insight into the characteristics of class 3 neighborhoods. Tracts with a higher share of class 3 neighborhoods had substantially higher shares of housing units in structures with 20 or more units in total (effect size of .47), lower homeownership rates (-.33), and lower shares of residents who commuted to work by car (-.15). Notably, class 3 neighborhoods also have the strongest association with household income (.11), suggesting that tracts with concentrations of class 3 neighborhoods have residents with higher-than-average incomes. These results suggest that class 3 represents mixed-use business and commercial areas with higher-than-average shares of

**Table 7. Percentage of classes by metro and location.**

| Metro | Class 1 | Class 2 | Class 3 | Class 4 | Class 5 |
|---|---|---|---|---|---|
| Atlanta | 24% | 46% | 10% | 18% | 3% |
| Boston | 31% | 34% | 6% | 16% | 13% |
| Chicago | 30% | 17% | 8% | 20% | 25% |
| Houston | 30% | 28% | 10% | 12% | 20% |
| Los Angeles | 19% | 14% | 12% | 23% | 33% |
| Central Cities | 21% | 6% | 14% | 11% | 48% |
| Suburban Cities | 34% | 11% | 9% | 22% | 24% |
| Urban Fringe | 21% | 44% | 8% | 17% | 10% |

**Table 8. Regression: Tract-level factors that predict the prevalence of each morphological class.**

| | Share of Blocks in Class 1 | Share of Blocks in Class 2 | Share of Blocks in Class 3 | Share of Blocks in Class 4 | Share of Blocks in Class 5 | Share of Blocks in Class 1 or Class 4 |
|---|---|---|---|---|---|---|
| (Intercept) | 0.16 *** | 0.38 *** | 0.00 | -0.23 *** | -0.33 *** | -0.02 |
| | (0.02) | (0.01) | (0.02) | (0.02) | (0.02) | (0.02) |
| Median Year Structure Built | 0.12 *** | 0.04 *** | 0.11 *** | 0.26 *** | -0.41 *** | 0.28 *** |
| | (0.01) | (0.01) | (0.01) | (0.01) | (0.01) | (0.01) |
| Population density | 0.33 *** | -0.64 *** | -0.11 *** | 0.18 *** | 0.27 *** | 0.41 *** |
| | (0.01) | (0.01) | (0.01) | (0.01) | (0.01) | (0.01) |
| Median Household Income | -0.07 *** | -0.02 * | 0.11 *** | -0.06 *** | 0.03 ** | -0.10 *** |
| | (0.01) | (0.01) | (0.01) | (0.01) | (0.01) | (0.01) |
| Non-Hispanic White (%) | -0.12 *** | 0.11 *** | -0.04 *** | 0.08 *** | -0.02 * | -0.05 *** |
| | (0.01) | (0.01) | (0.01) | (0.01) | (0.01) | (0.01) |
| Homeownership Rate | 0.24 *** | 0.02 * | -0.33 *** | 0.23 *** | -0.13 *** | 0.37 *** |
| | (0.02) | (0.01) | (0.01) | (0.02) | (0.01) | (0.01) |
| Commute by Car (%) | 0.11 *** | -0.03 *** | -0.15 *** | 0.08 *** | 0.00 | 0.15 *** |
| | (0.01) | (0.01) | (0.01) | (0.01) | (0.01) | (0.01) |
| Units in Structures with 20 + Units (%) | -0.18 *** | 0.03 *** | 0.47 *** | -0.08 *** | -0.18 *** | -0.22 *** |
| | (0.01) | (0.01) | (0.01) | (0.01) | (0.01) | (0.01) |
| Boston | 0.22 *** | -0.06 ** | -0.16 *** | 0.15 *** | -0.11 *** | 0.29 *** |
| | (0.03) | (0.02) | (0.03) | (0.03) | (0.03) | (0.03) |
| Chicago | 0.09 ** | -0.69 *** | 0.08 *** | 0.41 *** | 0.21 *** | 0.37 *** |
| | (0.03) | (0.02) | (0.02) | (0.03) | (0.02) | (0.03) |
| Houston | -0.04 | -0.38 *** | 0.14 *** | -0.34 *** | 0.51 *** | -0.27 *** |
| | (0.03) | (0.02) | (0.02) | (0.03) | (0.02) | (0.03) |
| Los Angeles | -0.54 *** | -0.50 *** | -0.03 | 0.49 *** | 0.62 *** | -0.13 *** |
| | (0.03) | (0.02) | (0.02) | (0.03) | (0.02) | (0.03) |
| N | 12072 | 12072 | 12072 | 12072 | 12072 | 12072 |
| R2 | 0.20 | 0.67 | 0.53 | 0.18 | 0.52 | 0.30 |

All continuous predictors and the outcome variable are mean-centered and scaled by 1 standard deviation

*** p < 0.001

** p < 0.01

* p < 0.05.

multifamily housing, rental housing, and multi-modal means of transit. The metropolitan dummy variables in the regression shown in Table 8 represent the average difference in the share of neighborhoods of each class relative to the reference category (Atlanta), holding other variables constant. We do not interpret these coefficients directly as they are used simply to control for variation in the prevalence of each class at the metropolitan level and largely substantiate the findings in Table 8.

At the opposite end of the spectrum are class 2 neighborhoods which, as discussed earlier, are characterized by low-density/high-variability development. The distribution of class 2 neighborhoods varies substantially, both across metro areas and sub-metropolitan contexts. For example, class 2 makes up 46% of neighborhoods in Atlanta but only 14% and 17% in Los Angeles and Chicago, respectively (see Table 3 and Fig 3). Similarly, class 2 is very common in the urban fringe (44% of neighborhoods), but uncommon in suburban cities (11%) and central cities (6%). The regression results also highlight that tracts with high shares of class 2 neighborhoods have exceedingly low population densities (effect size of -.64; see Table 8). Each of these

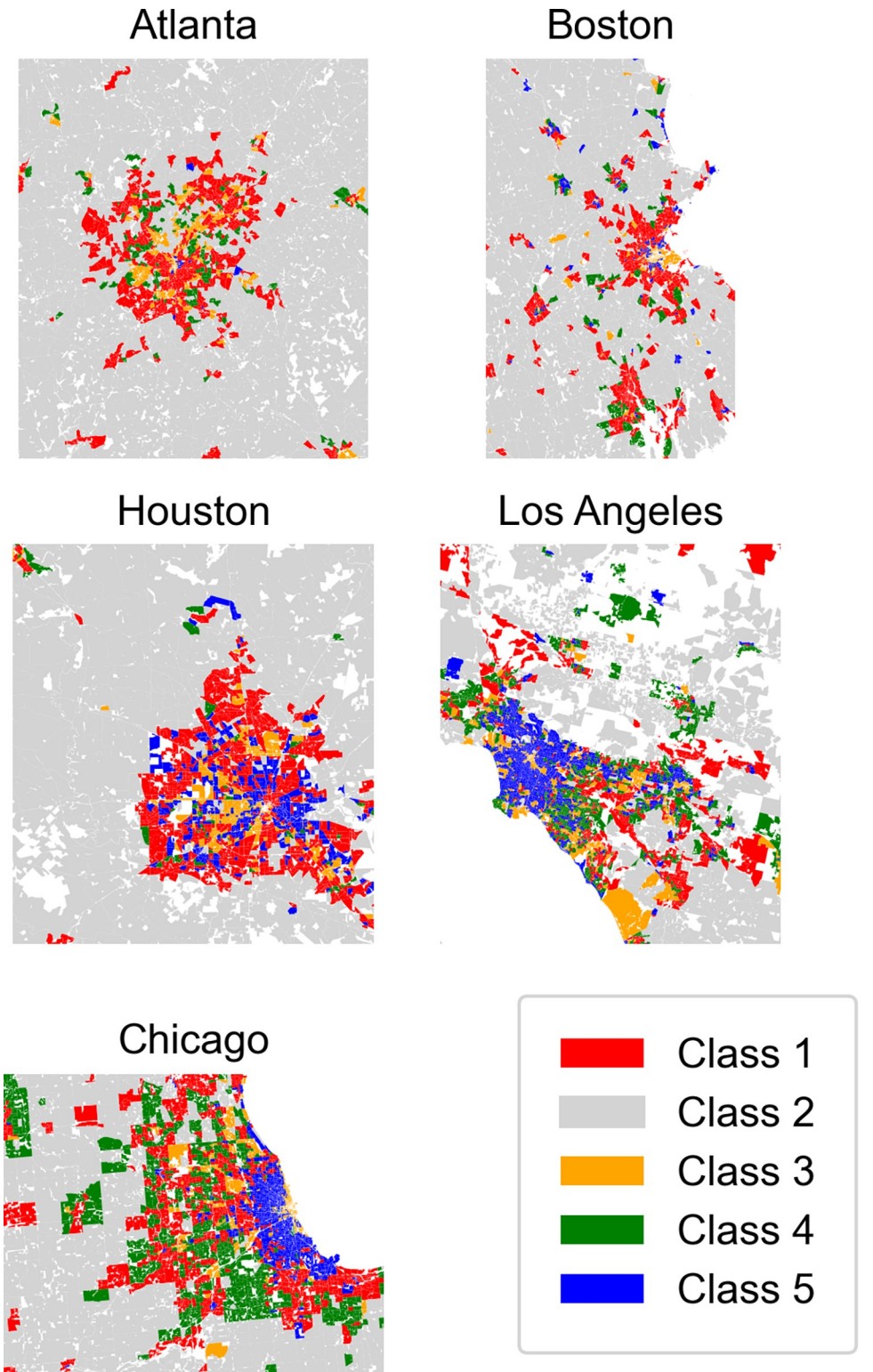

**Fig 3. Distribution of the five classes across the metropolitan landscape.**

statistics suggests that class 2 neighborhoods represent low-intensity development on the urban fringe. This conclusion is also supported by the fact that the concentration of class 2 neighborhoods has a significant but modest association with the share of non-Hispanic Whites (.11); counterintuitively, however, commuting by car has a small though statistically significant association with the prevalence of class 2 neighborhoods (-.03), the reason for which is unclear.

We now turn to a discussion of classes 1, 4, and 5. As we noted earlier, these three classes are relatively similar in their morphology: all three contain smaller, closely spaced (i.e., high-density) buildings. The main morphological differences between the three are A) that class 5 neighborhoods have greater variability in building size and shape than do classes 1 and 4, and B) that class 4 has larger buildings than class 1 (see Table 3). However, their distribution across space and their socioeconomic and demographic profiles differ in important ways. To illustrate this fact, we begin by discussing class 5 and how its physical morphology relates to tract-level socioeconomic and demographic characteristics that distinguish it from neighborhoods in classes 1 and 4.

As illustrated in Table 4, class 5 neighborhoods are most common in central cities (48% of neighborhoods) and least common on the urban fringe (10%). Classes 1 and 4, on the other hand, are more common in suburban cities (34% and 22% of neighborhoods, respectively) than in central cities (21% and 11%) or the urban fringe (21% and 17%). Class 5 thus likely represents older residential neighborhoods in dense urban centers, while classes 1 and 4 are primarily suburban neighborhoods. Thus, while class 5 is made up of small and densely spaced buildings, their location near central cities likely means these are some of the oldest residential neighborhoods, or "first suburbs," built before the dominance of subdivision regulations and zoning ordinances when more variability in housing forms (i.e. townhomes and row houses alongside single-family homes) was common [24]. The regression results at the tract level confirm these distinctions. For example, although tracts with high shares of class 1 neighborhoods have a small positive association with the median year of construction (.12), those with high shares of class 5 neighborhoods have a much larger, negative association (-.41); thus, tracts with newer housing are more likely to contain class 1 neighborhoods and less likely to contain class 5 neighborhoods.

Although classes 1 and 5 share some similarities, other characteristics of class 1 are indicative of suburban neighborhoods, while those of class 5 suggest they contain older urban neighborhoods. For example, tracts with high shares of class 1 and class 5 neighborhoods have high population densities (effect sizes of .33 and .27, respectively; see Table 8) and both have low percentages of multifamily structures (i.e., the share of units in structures with 20 or more units; -.18). However, whereas high concentrations of class 5 neighborhoods have a negative association with homeownership rates (-.13), homeownership is closely associated with the prevalence of class 1 neighborhoods (.24). Similarly, shares of commuting by car are not associated with class 5 neighborhoods but are common in class 1 (.11). These statistics, along with differences in the median year housing was built in each class, point to class 5 as older urban neighborhoods with a mix of owners and renters and class 1 as more recent suburban neighborhoods with high concentrations of homeowners.

We conclude by examining the socioeconomic and demographic characteristics of class 4 neighborhoods, paying particular attention to how they differ from those in class 1. As our earlier analysis of building morphology illustrated, class 4 neighborhoods have larger building footprints and lower density than in class 1. Once again, the demographic and socioeconomic characteristics provide insight into the social context for these differences. For example, in tracts with high shares of class 4 neighborhoods, the median structure was built more recently (effect size of .26; see Table 8) than in tracts with class 1 neighborhoods (.12). Similarly, tracts

with high shares of class 4 neighborhoods have lower population densities (.18 vs. .33). These statistics suggest that class 1 neighborhoods may represent earlier suburbs while class 4 neighborhoods represent more recent development; their morphology corresponds with the decade-by-decade increase in the average size of US homes that accompanied widespread suburbanization. That said, however, it is notable that the sign and magnitude of the coefficients are similar across the two models predicting the share of class 1 and class 4 neighborhoods, and that the r-squared in these models is substantially lower (.2 and .18, respectively) than for classes 2, 3, and 5 (.67, .53, and .52).

The low r-squared suggests, along with similarities in their morphological characteristics, suggest that class 1 and class 4 neighborhoods may not be distinct enough to warrant being considered separate types of neighborhood. To examine whether collapsing these two classes into a single neighborhood type led to changes in the regression results, we estimated a sixth regression model predicting the share of neighborhoods in either class 1 or 4. The results, shown in the last column in Table 8, provide some evidence that classes 1 and 4 represent similar neighborhood types. For example, after combining the two categories, the r-squared increases to .3 while the coefficients typically have the same sign as in the first and fourth models but are generally larger in magnitude.

## Discussion and conclusion

Neighborhood morphology–as represented by the size, shape, and placement of building footprints–provides a high-resolution means of measuring patterns of development across the urban landscape. In this paper, we examine whether neighborhood morphometrics at the census block level provide insight into spatial patterns of development and socioeconomic and demographic conditions across metropolitan and sub-metropolitan areas. We observe substantial differences in the size and placement of buildings across the five metropolitan areas, as well as across central cities, suburban cities, and the urban fringe. We also use unsupervised classification to develop a morphological typology of neighborhoods and examine variation in the prevalence of neighborhood types across urban space and its association with neighborhood-level socioeconomic and demographic conditions. Our cluster analysis reveals a set of five neighborhood types, including "first suburb" neighborhoods with modest and uniform housing size and placement; newer suburbs with larger but relatively uniform housing; older, high-density neighborhoods with highly varied housing; low-density neighborhoods with highly varied patterns of development; and neighborhoods with larger commercial or multi-family buildings. By comparing the prevalence of these neighborhood types across three metropolitan scales (urban, suburban, and urban fringe) and with tract-level socioeconomic and demographic data, we provide additional nuance regarding differences in the period of development, type of housing, characteristics of residents, and connection to employment opportunities across different neighborhood types. In doing so, we demonstrate a method of characterizing neighborhood morphology, detail a typology of U.S. neighborhoods across varying U.S. metros, and examine how different neighborhood morphologies align with variation in spatial and sociodemographic characteristics such as population density, prevalence of multifamily housing, income, race/ethnicity, homeownership, and commuting by car.

Beyond a typology of U.S. neighborhoods, the growing availability of building footprint data and an increasing number of statistical software programs for analyzing them [7,32] make possible a wide variety of analyses of neighborhood morphology that have the potential to advance geographic science in urban areas in important ways. Detailed data from the U.S. Census Bureau on neighborhood level conditions (e.g., the type and size of dwellings) are only available at the census block group level. However, block groups are often large, arbitrarily

delineated and contain a mixture of housing and neighborhood types. Building footprints and morphometrics derived from them provide a high-resolution option for distinguishing between different types of development at various spatial scales.

While it is beyond the scope of this paper to analyze all the ways physical morphology relates to tract-level socioeconomic and demographic characteristics, the association between neighborhood morphology and key socio-spatial characteristics indicates a number of significant applications of this method. Building footprint-derived estimates of neighborhood morphology provide an additional, high resolution means of analyzing patterns of urban development. As we illustrate, morphometrics capture variability in the layout of buildings and, in doing so, capture distinct morphological characteristics that reflect historical and contextual differences in development patterns across central cities, suburbs, and the urban fringe. Morphometrics may therefore be useful as primary or supplemental data inputs for efforts to examine and address a myriad of issues such as zoning and land use, housing supply and policy, residential segregation, neighborhood change, infrastructure investment, the development and operation of transit networks, historic preservation, and the coordination of regional development.

Future research could examine the causes of neighborhood morphology and its potential association with important societal outcomes. For example, scholars might use neighborhood morphology as the dependent variable in analyses of the impact of land use regulation, code enforcement actions, lending policy, and developer practices to understand how these policy and market factors shape the supply of housing and, as a result, the morphology of new neighborhoods. Similarly, scholars might use neighborhood morphology as the independent variable in analyses of residential segregation, economic mobility, or environmental vulnerability to understand how patterns of development shape access to opportunity or exposure to risk. As the availability of building footprints (or the aerial imagery used to derive them) increases, scholars could also examine temporal variation in development patterns and neighborhood morphology. This in turn could be used to examine physical patterns of neighborhood change (e.g., abandonment, infill, and upgrading) and socioeconomic or demographic patterns of neighborhood change (e.g., filtering, population loss, gentrification, etc.).

Future research might also address some of the limitations of the methods used here. For example, our method of unsupervised classification undoubtedly aggregates distinct neighborhoods into only a handful of neighborhood types. Scholars could use footprint-derived morphometrics and ground-truthed (parcel or zoning) data to distinguish between single-family and multifamily neighborhoods, manufactured home communities, and mixed-use developments. Future research could also explore alternative means of delineating neighborhood boundaries other than census blocks, including other census geographies, plat maps, or zoning districts. Additionally, morphological analysis might compress long, place-based histories into a geographic cross-section of the built environment. Thus, morphological analysis can be used to complement analyses of administrative, regulatory, and development data, thus opening multiple avenues of future research that can provide deeper insight into development patterns and economic or social phenomena.

## Author Contributions

**Conceptualization:** Noah J. Durst, Esther Sullivan.

**Data curation:** Noah J. Durst.

**Formal analysis:** Noah J. Durst.

**Funding acquisition:** Noah J. Durst, Esther Sullivan.

**Investigation:** Noah J. Durst.

**Methodology:** Noah J. Durst, Warren C. Jochem.

**Project administration:** Noah J. Durst.

**Resources:** Noah J. Durst.

**Supervision:** Noah J. Durst.

**Validation:** Noah J. Durst.

**Visualization:** Noah J. Durst.

**Writing – original draft:** Noah J. Durst, Esther Sullivan, Warren C. Jochem.

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
