## [Decision Letter · Decision Letter 0]

7 Sep 2023

PONE-D-23-21488The spatial and social correlates of neighborhood morphology: Evidence from building footprints in five U.S. metropolitan areasPLOS ONE

Dear Dr. Durst,

Thank you for submitting your manuscript to PLOS ONE. After careful consideration, we feel that it has merit but does not fully meet PLOS ONE’s publication criteria as it currently stands. Therefore, we invite you to submit a revised version of the manuscript that addresses the points raised during the review process.

We look forward to receiving your revised manuscript.

Kind regards,

Gang Xu, Ph.D.

Academic Editor

PLOS ONE

4. We note that Figures 1 and 2 in your submission contain map images which may be copyrighted. All PLOS content is published under the Creative Commons Attribution License (CC BY 4.0), which means that the manuscript, images, and Supporting Information files will be freely available online, and any third party is permitted to access, download, copy, distribute, and use these materials in any way, even commercially, with proper attribution. For these reasons, we cannot publish previously copyrighted maps or satellite images created using proprietary data, such as Google software (Google Maps, Street View, and Earth). For more information, see our copyright guidelines: http://journals.plos.org/plosone/s/licenses-and-copyright.

1. You may seek permission from the original copyright holder of Figures 1 and 2 to publish the content specifically under the CC BY 4.0 license. 

Additional Editor Comments:

The reviewers recommend reconsideration of your manuscript following major revision. I invite you to resubmit your manuscript after addressing their comments.

Reviewers' comments:

Reviewer's Responses to Questions

**Comments to the Author**

1. Is the manuscript technically sound, and do the data support the conclusions?

Reviewer #1: Yes

Reviewer #2: Partly

2. Has the statistical analysis been performed appropriately and rigorously? 

Reviewer #1: Yes

Reviewer #2: Yes

3. Have the authors made all data underlying the findings in their manuscript fully available?

Reviewer #1: Yes

Reviewer #2: No

4. Is the manuscript presented in an intelligible fashion and written in standard English?

Reviewer #1: Yes

Reviewer #2: Yes

5. Review Comments to the Author

Reviewer #1: This paper examined the different neighborhood morphologies at the census tract level in U.S. metropolitan areas and their relationship to demographic and socioeconomic characteristics. This paper has many important issues that need to be revised and supplemented before it can be carefully considered for publication.

1. The Abstract and Discussion sections of this paper need to be improved.

2. The Methods section of the paper should be supplemented with necessary formulas and technical route descriptions.

3. There are only two Figures at the end of the document, and the title is missing? Their resolution is low, and not standardized and clear enough, and the necessary labels are lacking. Figure 2 should add study unit boundaries. According to the context, are the labels in Figure 1 and Figure 2 incorrect?

4. It will be easier to understand if the morphological characteristics indicators and demographic and socioeconomic characteristics indicators used in this paper are listed and explained separately in tables.

5. In this paper, it is necessary to explain how to accurately divide the range of "central cities, suburban cities, and the urban fringe".

6. Please explain the meaning of the regression results for different metropolitan areas in Table 5.

7. At the end of this paper, the analysis of the regression results of the five distinct neighborhood types and demographic and socioeconomic characteristics is too simple, and the results in different metropolitan areas should be discussed and analyzed.

Reviewer #2: This study investigates the neighborhood morphologies of five U.S. metropolitan areas (MAs) by revealing their morphological features based on morphometrics, identifying their patterns based on unsupervised classification, and constructing their spatial and social correlates based on Ordinary Least Squares regression. Generally, the manuscript is clearly written, the research method is relatively novel, and the research perspective is unique. However, I think that there also exist some main issues with this manuscript. First, the research significance and contribution are unclear. Second, the innovation and improvement of this study compared to Jochem and Tatem (2021)’s research are uncertain. Finally, the literature review is not intimately related to the topic of this article. Detailed comments are listed as follows:

1. I think that the current abstract lacks key information about the research significance and contribution. Excessive introduction of the method background and processes can be further streamlined. In addition, the core conclusion of this paper should be briefly mentioned as a response to the research significance and contribution.

2. The introduction explains the research meaning of urban morphology. As its branch and the main research object of this study, the research significance of neighborhood morphology is not clearly illustrated. Specifically, why should we investigate neighborhood morphology? What can we learn from neighborhood morphology that other studies on urban morphology cannot provide? What is the practical meaning of investigating neighborhood morphology?

3. Page 4, Paragraph 1. “We answer three primary research questions: … Do neighborhoods with distinct building morphologies differ in regard to their key socio-demographic characteristics?” Following the 2nd comment, I have no idea how to propose the three primary research questions and why we need to answer them due to the insufficient introduction of research significance and social background. What are the contributions to society by answering them?

4. I think that the vague research topic and unclear research significance of this article lead to a weak literature review. Literature review should not focus on a single disciplinary, but should be organized based on the research topic as many studies are currently interdisciplinary. The concluded limitation of current research is that “the existing morphological focus on distinguishing urbanized vs. non-urbanized development often fails to capture the complex, often polycentric reality of urbanization”. Is this study working on urban polycentricity or urbanization? About polycentric reality, at least studies on human mobility can identify the pattern of urban polycentricity. Why not mention them? Can the method in this paper identify and capture polycentric reality?

5. I noticed that this study draws on and applies the method of Jochem and Tatem (2021). What are the main differences between this study and their research, apart from different study areas? What are the innovations and contributions of this study? What improvements have been made in this study compared to their research?

6. Why use neighborhood morphometrics in the foot package of R? What is its advantage? Does this package include all morphometrics? If not, why not use other morphological metrics?

7. Page 8, Paragraph 2. “For more information on these measurements, see Jochem and Tatem (2021)” This is too casual. Do we still need to look for their article to interpret your results? I think a simple table containing basic information about these metrics is necessary, such as their full names, abbreviations, description, and calculation formulas.

8. Is the comparison across classification algorithms appropriate by only using a sample of census blocks? Are the results (e.g., the characteristics of all classes) of unsupervised classification based on this sample similar to that based on the whole research area? Is the silhouette scores an appropriate basis for comparing algorithms as the classification results with the highest scores do not provide much insight into variation within these classes. Are there any other evaluation bases? Where are the comparison results of these algorithms?

9. Why is the median distance between buildings in the old MA of Boston larger than that in the post-car MAs such as Houston and Los Angeles? Why does the MA of Los Angeles with the most dense arrangement of buildings have the largest size of building footprints? Are these results contradictory? Could you please explain these?

10. Why are there no median morphometrics of three types of locations (central cities, suburban cities, and urban fringe) by MA? Do these MAs have the similar characteristics of building footprints at this location scale? Similarly, what about the classification results for each MA?

11. The resolution of Figure 1 is too low.

12. What are the reasons for the lower commuting by car in Class 2?

13. The overall goodness of fit for Class 1 and 3 is too low. Can the Ordinary Least Square regression really represent the relationships between the shares of neighborhoods and socioeconomic and demographic indicators?

6. PLOS authors have the option to publish the peer review history of their article (what does this mean?). If published, this will include your full peer review and any attached files.

Reviewer #1: No

Reviewer #2: No

---

## [Author Response · Author response to Decision Letter 0]

22 Oct 2023

RESPONSE: The manuscript has been updated accordingly.

RESPONSE: Accompanying data and code will be made available via the Inter-university Consortium for Political and Social Research (ICPSR) at the time of acceptance of the manuscript.

4. We note that Figures 1 and 2 in your submission contain map images which may be copyrighted. All PLOS content is published under the Creative Commons Attribution License (CC BY 4.0), which means that the manuscript, images, and Supporting Information files will be freely available online, and any third party is permitted to access, download, copy, distribute, and use these materials in any way, even commercially, with proper attribution. For these reasons, we cannot publish previously copyrighted maps or satellite images created using proprietary data, such as Google software (Google Maps, Street View, and Earth). For more information, see our copyright guidelines: http://journals.plos.org/plosone/s/licenses-and-copyright.

1. You may seek permission from the original copyright holder of Figures 1 and 2 to publish the content specifically under the CC BY 4.0 license. 

RESPONSE: These images are not under copyright. We generated them using the Python programming language using publicly available U.S. Census Bureau boundary files. We grant permission for publication.

 Additional Editor Comments:

 The reviewers recommend reconsideration of your manuscript following major revision. I invite you to resubmit your manuscript after addressing their comments.

 1. Is the manuscript technically sound, and do the data support the conclusions?

Reviewer #1: Yes

Reviewer #2: Partly

RESPONSE: Thank you for this feedback. We have considerably expanded the methods and data section to address this concern.

2. Has the statistical analysis been performed appropriately and rigorously?

Reviewer #1: Yes

Reviewer #2: Yes

3. Have the authors made all data underlying the findings in their manuscript fully available?

Reviewer #1: Yes

Reviewer #2: No

RESPONSE: Block- and tract-level morphometrics, as well as results of the principal components and cluster analyses, will be made publicly available via the Inter-university Consortium for Political and Social Research (ICPSR) at the time of acceptance of the manuscript. The building footprint data used to create these morphometrics are publicly available from Microsoft (https://github.com/microsoft/USBuildingFootprints). Block, place, and metropolitan area shapefiles and corresponding data are publicly available from the National Historical Geographic Information System (NHGIS; www.nhgis.org). 

4. Is the manuscript presented in an intelligible fashion and written in standard English?

Reviewer #1: Yes

Reviewer #2: Yes

5. Review Comments to the Author

Reviewer #1: This paper examined the different neighborhood morphologies at the census tract level in U.S. metropolitan areas and their relationship to demographic and socioeconomic characteristics. This paper has many important issues that need to be revised and supplemented before it can be carefully considered for publication.

 1. The Abstract and Discussion sections of this paper need to be improved.

RESPONSE: The Abstract and Discussion have been revised and improved in light of both reviewers' helpful comments. The Abstract is now more focused, direct, and better articulates our primary contribution and the Discussion section has been expanded to reflect the addition of a number of new tables and expanded discussion of the results. The Discussion & Conclusion section has also been expanded to include a more focused discussion of the significance of our method to zoning & land use, transportation planning, housing affordability, infrastructure investment and environmental planning, historic preservation, and regional development strategies and associated research in these areas. See major revisions on PG 2 and PGs 33-34.

 2. The Methods section of the paper should be supplemented with necessary formulas and technical route descriptions.

RESPONSE: Thank you for this suggestion. We have considerably expanded the discussion of the methods used in the study. As described below, this includes expanded discussion and detail regarding the measurement of each morphometric variable at the building and census block level (see Table 1), additional details regarding the testing and performance evaluation for different classification algorithms (see Table 2), and the inclusion of the regression equation and accompanying expanded discussion (see PG 15-16).

 3. There are only two Figures at the end of the document, and the title is missing? Their resolution is low, and not standardized and clear enough, and the necessary labels are lacking. Figure 2 should add study unit boundaries. According to the context, are the labels in Figure 1 and Figure 2 incorrect?

RESPONSE: Thank you for pointing this out. The figure captions are included in the text. We have prepared high-resolution, publication-ready versions of all images. We have confirmed that all images have correct labels and legends. We have not added study unit boundaries to Figure 2, as the boundaries of the study area are delineated in white shading. 

 4. It will be easier to understand if the morphological characteristics indicators and demographic and socioeconomic characteristics indicators used in this paper are listed and explained separately in tables.

RESPONSE: We have added Table 1, which explains the name of each variable and the building- and block-level measurements used.

 5. In this paper, it is necessary to explain how to accurately divide the range of "central cities, suburban cities, and the urban fringe".

RESPONSE: Thank you. We have provided additional detail regarding how we used U.S. Census Bureau records to distinguish between central cities, suburban cities, and the urban fringe (see PGs 11-12).

 6. Please explain the meaning of the regression results for different metropolitan areas in Table 5.

RESPONSE: Thank you. We have added a discussion clarifying that the metropolitan dummy variables in the regression represent the average difference in the share of neighborhoods of each class relative to the reference category (Atlanta), holding other factors constant. We also note that we do not interpret these coefficients directly as they are used simply to control for variation in the prevalence of each class at the metropolitan level and largely substantiate the descriptive statistics presented earlier in the results section. 

 7. At the end of this paper, the analysis of the regression results of the five distinct neighborhood types and demographic and socioeconomic characteristics is too simple, and the results in different metropolitan areas should be discussed and analyzed. 

RESPONSE: As we note above, we chose not to provide an expanded discussion of the metropolitan dummy variables. These findings are somewhat awkward to discuss because they represent the average difference in the share of neighborhoods of each class relative to the reference category (Atlanta) and holding all other variables constant. Moreover, a more straightforward discussion of descriptive statistics illustrating variation of the prevalence of each class across metro areas is already a central part of the analysis within that section. As noted above, we have added a brief discussion to clarify for the reader the interpretation of the dummy variables. 

Reviewer #2: This study investigates the neighborhood morphologies of five U.S. metropolitan areas (MAs) by revealing their morphological features based on morphometrics, identifying their patterns based on unsupervised classification, and constructing their spatial and social correlates based on Ordinary Least Squares regression. Generally, the manuscript is clearly written, the research method is relatively novel, and the research perspective is unique. However, I think that there also exist some main issues with this manuscript. First, the research significance and contribution are unclear. Second, the innovation and improvement of this study compared to Jochem and Tatem (2021)’s research are uncertain. Finally, the literature review is not intimately related to the topic of this article. Detailed comments are listed as follows:

RESPONSE: These are helpful comments and we address each in turn below.

 1. I think that the current abstract lacks key information about the research significance and contribution. Excessive introduction of the method background and processes can be further streamlined. In addition, the core conclusion of this paper should be briefly mentioned as a response to the research significance and contribution.

RESPONSE: The Abstract has been revised to eliminate excessive methodological details and articulate our key contribution & the significance/applicability of this method to research and policy related zoning & land use, transportation planning, housing affordability, infrastructure investment and environmental planning, historic preservation, and regional development strategies and associated research in these areas. See revisions on PG 2.

 2. The introduction explains the research meaning of urban morphology. As its branch and the main research object of this study, the research significance of neighborhood morphology is not clearly illustrated. Specifically, why should we investigate neighborhood morphology? What can we learn from neighborhood morphology that other studies on urban morphology cannot provide? What is the practical meaning of investigating neighborhood morphology?

RESPONSE: Thank you for these comments. First, we have streamlined and removed some references to the urban morphology literature to better highlight our primary contribution to investigating neighborhood morphology in its own right. Second, we make clear that the new data, tools, and techniques that we reference mean scholars are not limited to small case studies which are/were so common in urban morphology studies. Our study tests our method of characterizing neighborhood morphology across five of the ten largest U.S metropolitan areas and develops a typology of U.S. neighborhoods based on morphologic characteristics. Third, we revised our paper to better articulate our focus on neighborhoods and the importance of investigating neighborhood morphology in its own right, and to distinguish this from the broader study of urban morphology. We include reference to a broad interdisciplinary literature that has established that neighborhoods matter for a range of social, political and ecological outcomes. Our purpose in this paper is to explore how neighborhood morphology matters in these processes and how it maps on to existing spatial and sociodemographic characteristics of places. We now explore several potential practical applications of this method for classifying neighborhood morphology in the conclusion (PG 34-35). These revisions can be found throughout the Introduction and Literature Review, and are further detailed in response to the comments below. Revisions begin at the outset of the manuscript on PGs 3-4 and continue throughout the Literature Review. 

 3. Page 4, Paragraph 1. “We answer three primary research questions: … Do neighborhoods with distinct building morphologies differ in regard to their key socio-demographic characteristics?” Following the 2nd comment, I have no idea how to propose the three primary research questions and why we need to answer them due to the insufficient introduction of research significance and social background. What are the contributions to society by answering them?

RESPONSE: Thank you for this comment. As we note above, we have considerably revised the discussion of the literature to situate our research questions within it and to clarify their contribution to the literature. The Discussion & Conclusion section has also been expanded to include a more focused discussion of the significance of our method to zoning & land use, transportation planning, housing affordability, infrastructure investment and environmental planning, historic preservation, and regional development strategies and associated research in these areas

 4. I think that the vague research topic and unclear research significance of this article lead to a weak literature review. Literature review should not focus on a single disciplinary, but should be organized based on the research topic as many studies are currently interdisciplinary. The concluded limitation of current research is that “the existing morphological focus on distinguishing urbanized vs. non-urbanized development often fails to capture the complex, often polycentric reality of urbanization”. Is this study working on urban polycentricity or urbanization? About polycentric reality, at least studies on human mobility can identify the pattern of urban polycentricity. Why not mention them? Can the method in this paper identify and capture polycentric reality?

RESPONSE: We appreciate this comment and the opportunity to frame our analysis in terms of its clear implications beyond a single discipline, notably for urban studies and planning as well. In our study, we focus on the issue of neighborhood morphology, an under-studied aspect of urban morphological research. In doing so, we examine the spatial distribution and patterning of neighborhood morphology and their association with sociodemographic factors. As we now describe, the analysis of neighborhood morphologies offers a more nuanced understanding of the built environment in urban areas by looking at spatial distribution of these neighborhood classes and associated sociodemographic conditions. In conclusion we better examine the implications for urban studies and planning, noting the use of our method for examining how different neighborhood morphologies might be associated with different development patterns, housing types, infrastructure and economic characteristics. We have removed all references to polycentrism because we recognize that this term is used differently across fields (i.e. from planning to civil engineering). We agree that for PLOS ONEs interdisciplinary readership it is better to frame the research in terms of its interdisciplinary significance, which we have done in the revised Introduction. 

 5. I noticed that this study draws on and applies the method of Jochem and Tatem (2021). What are the main differences between this study and their research, apart from different study areas? What are the innovations and contributions of this study? What improvements have been made in this study compared to their research?

RESPONSE: We agree that our study does draw on some of the methods demonstrated in Jochem and Tatem (2021), specifically the use of the R package foot to compute a range of morphometrics of 2D building footprints and the use of unsupervised clustering algorithms. However, they demonstrate the generation and use of “gridded” (raster) morphometric datasets and they state that their study was primarily a demonstration of programming tools with less emphasis given to the interpretation of the results beyond a comparison with existing typologies. Our study contributes a more nuanced interpretation of the morphometrics themselves before building the typologies and further examining the distribution of the neighborhood types across metro areas. We also choose to work with Census geographies which enable our study to examine the association of sociodemographic characteristics with neighborhood typologies. As noted in our revised Discussed, we demonstrate how researchers can leverage morphological analyses to study land use, development and the impact of other policies.

 6. Why use neighborhood morphometrics in the foot package of R? What is its advantage? Does this package include all morphometrics? If not, why not use other morphological metrics?

RESPONSE: Thank you for this comment. While other tools could be used, we chose the foot package for its ease of use in our workflow and because it provides all the metrics we needed. We chose a subset of potential morphological measurements which we anticipated would show variation among buildings and are readily interpretable. We provide an expanded discussion of and justification for our use of the foot package and the selected morphometrics as well as their building- and block-level measurements, PGs 8-9.

 7. Page 8, Paragraph 2. “For more information on these measurements, see Jochem and Tatem (2021)” This is too casual. Do we still need to look for their article to interpret your results? I think a simple table containing basic information about these metrics is necessary, such as their full names, abbreviations, description, and calculation formulas.

RESPONSE: Thank you. We have provided a new table (Table 1) and accompanying text (PG 9) summarizing the building- and block-level measurements used to calculate each morphometric. 

 8. Is the comparison across classification algorithms appropriate by only using a sample of census blocks? Are the results (e.g., the characteristics of all classes) of unsupervised classification based on this sample similar to that based on the whole research area? Is the silhouette scores an appropriate basis for comparing algorithms as the classification results with the highest scores do not provide much insight into variation within these classes. Are there any other evaluation bases? Where are the comparison results of these algorithms? 

RESPONSE: Thank you for this comment. We have provided a more detailed discussion of the model evaluation criteria, including the silhouette score and estimates of over-segmentation for all models is now shown in Table 2. To illustrate over-segmentation in some models, we present in parentheses the number of classes containing fewer than 1% of observations. As the accompanying text on pages 12-13 illustrates, these statistics point toward the K-means and GMM models as the having the best performance. We choose K-means given its frequent use in the literature. We then present an elbow plot (Figure 1) for the K-means models with between 2 and 10 clusters, see PGs 12-13

 9. Why is the median distance between buildings in the old MA of Boston larger than that in the post-car MAs such as Houston and Los Angeles? Why does the MA of Los Angeles with the most dense arrangement of buildings have the largest size of building footprints? Are these results contradictory? Could you please explain these?

RESPONSE: Thank you for these helpful questions. We agree that these are somewhat counter-intuitive findings. We have added additional discussion of these results, which are driven in large part by variation in development patterns across locations within each metro area. We have added a new table (Table 5) and accompanying discussion (see response below) which explores this variation, see PG 17 for discussion and additional references.

 10. Why are there no median morphometrics of three types of locations (central cities, suburban cities, and urban fringe) by MA? Do these MAs have the similar characteristics of building footprints at this location scale? Similarly, what about the classification results for each MA?

RESPONSE: Thank you for this suggestion. We have added a new table of descriptive statistics to illustrate variation in the morphology of neighborhoods within each metro at the three spatial scales (see Table 5). The revised discussion that accompanies this table adds additional insight into some of the questions raised above. 

 11. The resolution of Figure 1 is too low.

RESPONSE: Thank you. The final version of the Figure is 600dpi.

 12. What are the reasons for the lower commuting by car in Class 2?

RESPONSE: Thank you for pointing this out. In the process of examining the cause of this issue, we identified and resolved an error that inadvertently led to the removal of a sample of census tracts from the regression analysis. The new regression results are revised to reflect this. In general, the changes are minor and do not change the overall conclusions. This relationship is one of the few that were substantively changed. This substantially attenuated in the revised regression results (the coefficient has declined from -.1 to -.03). We note in the text that the reason for this negative relationship is unclear, though it is small. 

 13. The overall goodness of fit for Class 1 and 3 is too low. Can the Ordinary Least Square regression really represent the relationships between the shares of neighborhoods and socioeconomic and demographic indicators?

RESPONSE: Thank you for pointing this out. The low r-square for classes 1 and 4 persist in the revised regression analysis. We note this limitation and provide an expanded discussion of two potential explanations. We also conduct an additional regression analysis after collapsing classes 1 and 4. As the results illustrate, the r-squared and the magnitude of the coefficients increase. This indicates that classes 1 and 4 may represent similar morphological classes with comparable socioeconomic and demographic profiles.

---

## [Decision Letter · Decision Letter 1]

26 Dec 2023

PONE-D-23-21488R1The spatial and social correlates of neighborhood morphology: Evidence from building footprints in five U.S. metropolitan areasPLOS ONE

Dear Dr. Durst,

Thank you for submitting your manuscript to PLOS ONE. After careful consideration, we feel that it has merit but does not fully meet PLOS ONE’s publication criteria as it currently stands. Therefore, we invite you to submit a revised version of the manuscript that addresses the points raised during the review process.

We look forward to receiving your revised manuscript.

Kind regards,

Gang Xu, Ph.D.

Academic Editor

PLOS ONE

Journal Requirements:

Additional Editor Comments:

Thanks for your revision. As one of the reviewer providing further comments, I would like to invite you resubmit your paper in a minor revision.

Reviewers' comments:

Reviewer's Responses to Questions

**Comments to the Author**

1. If the authors have adequately addressed your comments raised in a previous round of review and you feel that this manuscript is now acceptable for publication, you may indicate that here to bypass the “Comments to the Author” section, enter your conflict of interest statement in the “Confidential to Editor” section, and submit your "Accept" recommendation.

Reviewer #2: (No Response)

2. Is the manuscript technically sound, and do the data support the conclusions?

Reviewer #2: Yes

3. Has the statistical analysis been performed appropriately and rigorously? 

Reviewer #2: Yes

4. Have the authors made all data underlying the findings in their manuscript fully available?

Reviewer #2: Yes

5. Is the manuscript presented in an intelligible fashion and written in standard English?

Reviewer #2: Yes

6. Review Comments to the Author

Reviewer #2: The authors carefully and meticulously responded to my previous concerns, and made targeted revisions to the article. The completeness of the whole content in this manuscript has been further improved. However, I think that the current abstract and introduction sections need further revisions to enhance their readability and comprehensibility.

1. The abstract section should at least provide some key information, such as the significance of your research field, research gap (what problems have not been solved, which could be your motivation and draws out your research questions), and some main results and findings (to support your conclusion and implications). Without these information, it is difficult to understand your methods and procedures (what are their purposes?) as well as your conclusion and implications (where do they come from?). In addition, there are too many “we do” sentences. I think some of them can be replaced by the description of your main results and findings.

2. The introduction section needs further enhancement to be more informative, more logical and better organized. For example, the 1st paragraph in the background section are redundant and repetitive (as the 1st paragraph of the introduction has the similar gist). The proposal of three research questions is abrupt, lacking sufficient groundwork of research background, literature review, and research gaps (some of which I can only find later in the text). In addition, I think you should emphasize your research contributions more plainly, especially the progress you have made compared to Jochem and Tatem (2021)’s work and other previous studies.

7. PLOS authors have the option to publish the peer review history of their article (what does this mean?). If published, this will include your full peer review and any attached files.

Reviewer #2: No

---

## [Author Response · Author response to Decision Letter 1]

28 Jan 2024

Thank you for the very helpful feedback on our manuscript. Below we past the reviewer comments (prefaced by the word RECOMMENDATION) and respond to each substantive recommendation (prefaced by the word RESPONSE).

RECOMMENDATION: 

Reviewer #2: The authors carefully and meticulously responded to my previous concerns, and made targeted revisions to the article. The completeness of the whole content in this manuscript has been further improved. However, I think that the current abstract and introduction sections need further revisions to enhance their readability and comprehensibility.

1. The abstract section should at least provide some key information, such as the significance of your research field, research gap (what problems have not been solved, which could be your motivation and draws out your research questions), and some main results and findings (to support your conclusion and implications). Without these information, it is difficult to understand your methods and procedures (what are their purposes?) as well as your conclusion and implications (where do they come from?). In addition, there are too many “we do” sentences. I think some of them can be replaced by the description of your main results and findings.

RESPONSE: 

Thank you for these helpful recommendations. We have revised the abstract to reduce the methodological discussion and incorporate a discussion of the research gap/motivation as well as main results. We have also removed multiple instances of the use of “we.” The revised abstract is provided here as well as in the text:

“Recent advances in quantitative tools for examining urban morphology enable the development of morphometrics that can characterize the size, shape, and placement of buildings; the relationships between them; and their association with broader patterns of development. Although these methods have the potential to provide substantial insight into the ways in which neighborhood morphology shapes the socioeconomic and demographic characteristics of neighborhoods and communities, this question is largely unexplored. Using building footprints in five of the ten largest U.S. metropolitan areas (Atlanta, Boston, Chicago, Houston, and Los Angeles) and the open-source R package, foot, we examine how neighborhood morphology differs across U.S. metropolitan areas and across the urban-exurban landscape. Principal components analysis, unsupervised classification (K-means), and Ordinary Least Squares regression analysis are used to develop a morphological typology of neighborhoods and to examine its association with the spatial, socioeconomic, and demographic characteristics of census tracts. Our findings illustrate substantial variation in the morphology of neighborhoods, both across the five metropolitan areas as well as between central cities, suburbs, and the urban fringe within each metropolitan area. We identify five different types of neighborhoods indicative of different stages of development and distributed unevenly across the urban landscape: these include low-density neighborhoods on the urban fringe; mixed use and high-density residential areas in central cities; and uniform residential neighborhoods in suburban cities. Results from regression analysis illustrate that the prevalence of each of these forms is closely associated with variation in socioeconomic and demographic characteristics such as population density, the prevalence of multifamily housing, and income, race/ethnicity, homeownership, and commuting by car. We conclude by discussing the implications of our findings and suggesting avenues for future research on neighborhood morphology, including ways that it might provide insight into issues such as zoning and land use, housing policy, and residential segregation.”

RECOMMENDATION: 

2. The introduction section needs further enhancement to be more informative, more logical and better organized. For example, the 1st paragraph in the background section are redundant and repetitive (as the 1st paragraph of the introduction has the similar gist). The proposal of three research questions is abrupt, lacking sufficient groundwork of research background, literature review, and research gaps (some of which I can only find later in the text). In addition, I think you should emphasize your research contributions more plainly, especially the progress you have made compared to Jochem and Tatem (2021)’s work and other previous studies.

• the 1st paragraph in the background section are redundant and repetitive

• proposal of three research questions is abrupt, lacking sufficient groundwork of research background, literature review, and research gaps (some of which I can only find later in the text)

• I think you should emphasize your research contributions more plainly, especially the progress you have made compared to Jochem and Tatem (2021)’s work and other previous studies

RESPONSE: 

Thank you for these helpful recommendations. We agree that emphasizing the paper’s contribution and how it differs from previous studies is important. We also that additional foregrounding of the importance of neighborhoods in social research was important for clarifying the motivation of the research and the gap in the existing literature prior to articulating the research question. These following two paragraphs in the introduction now address both of these issues:

“New data, tools, and techniques mean researchers are not limited to small case studies which have been common in urban morphology studies. Recent research using building footprints has used morphological analysis to characterize patterns of development at the neighborhood level [7]. For example, Jochem and Tatem use publicly available spatial datasets of building footprints to define their constituent elements (size, shape, and placement of structures) in England, Scotland and Wales and to examine the extent to which typologies of neighborhoods derived from unsupervised classification using building footprint morphometrics align with census-defined classifications for rural and urban areas of various types [7]. 

We adapt and extend this analysis to the U.S. context to analyze the dimensions and distribution of development inscribed in the morphology of neighborhoods in five of the ten largest U.S metropolitan areas and to develop a typology of U.S. neighborhoods based on their morphological characteristics. In doing so, we combine the tools of urban morphology with the theoretical contributions from a vast literature in urban studies, sociology, and planning that has explored how neighborhoods are a key mechanism that structures ecological, political and social outcomes in metro regions . Distinct types of neighborhoods (e.g., suburban enclaves, urban cores, rural districts) vary markedly in the characteristics of their population and the opportunities they provide [12–14]. Little is known, however, about whether the morphological characteristics measured by building footprints align with these pre-existing conceptual understandings of neighborhoods and the characteristics of residents in them. We address this gap in this study by answering three primary research questions: Can neighborhood-level estimates of building morphology be used to create a useful typology of U.S. neighborhoods that maps onto conceptual understandings of urban form? How does neighborhood morphology vary across the country and across central cities, suburban areas, and the urban fringe? Do neighborhoods with distinct building morphologies differ in regard to key socio-demographic characteristics?”

We appreciate the feedback regarding the short paragraph at the start of the Background section. We have shortened that discussion to prevent duplication.

---

## [Decision Letter · Decision Letter 2]

19 Feb 2024

The spatial and social correlates of neighborhood morphology: Evidence from building footprints in five U.S. metropolitan areas

PONE-D-23-21488R2

Dear Dr. Durst,

We’re pleased to inform you that your manuscript has been judged scientifically suitable for publication and will be formally accepted for publication once it meets all outstanding technical requirements.

Kind regards,

Gang Xu, Ph.D.

Academic Editor

PLOS ONE

Additional Editor Comments (optional):

Thank you for your revison. The minor issues raised by the Reviewer #2 could be revised during the proof reading.

Reviewers' comments:

Reviewer's Responses to Questions

**Comments to the Author**

1. If the authors have adequately addressed your comments raised in a previous round of review and you feel that this manuscript is now acceptable for publication, you may indicate that here to bypass the “Comments to the Author” section, enter your conflict of interest statement in the “Confidential to Editor” section, and submit your "Accept" recommendation.

Reviewer #2: All comments have been addressed

2. Is the manuscript technically sound, and do the data support the conclusions?

Reviewer #2: Yes

3. Has the statistical analysis been performed appropriately and rigorously? 

Reviewer #2: Yes

4. Have the authors made all data underlying the findings in their manuscript fully available?

Reviewer #2: Yes

5. Is the manuscript presented in an intelligible fashion and written in standard English?

Reviewer #2: Yes

6. Review Comments to the Author

Reviewer #2: In this round of revision, the authors have generally responded to my concerns. Before final acceptance, however, I think that minor modifications still need to be made.

1. The organizational logic of the first paragraph in the introduction is still a little unclear. I still have no clue about the semantic function of the first sentence (“A decades-long shift in how geographers and planners analyze urban form has emphasized how bottom-up and uncoordinated local decision-making gives rise to large-scale, coordinated, morphological patterns that define the size and shape of cities in predictable ways”) and its connection with the following contents. In addition, the statement, “Morphological understanding of urban spatial organization and evolution can identify underlying mechanisms and characteristics of urban development, to better plan for and manage increasingly complex urban areas”, seems to be the significance of research on urban morphology whether for early visual observation or later quantitative characterization. It seems inappropriate to appear here.

2. The first sentence (“A wide body of literature in the geographic sciences has focused hhas sought to use morphological analysis to examine urban phenomena, including the variegated character of urban development and neighborhood-scale distinctions between settlement types” ?) in the background should be revised.

7. PLOS authors have the option to publish the peer review history of their article (what does this mean?). If published, this will include your full peer review and any attached files.

Reviewer #2: No

---

## [Editor Report · Acceptance letter]

6 Mar 2024

PONE-D-23-21488R2 

PLOS ONE

Dear Dr. Durst, 

I'm pleased to inform you that your manuscript has been deemed suitable for publication in PLOS ONE. Congratulations! Your manuscript is now being handed over to our production team.

Kind regards, 

on behalf of

Dr. Gang Xu 

Academic Editor

PLOS ONE